# From Chemical Composition to Antiproliferative Effects Through In Vitro Studies: Honey, an Ancient and Modern Hot Topic Remedy

**DOI:** 10.3390/nu17091595

**Published:** 2025-05-06

**Authors:** Alexandru Nan, Victor Dumitrascu, Corina Flangea, Gabi Dumitrescu, Daniela Puscasiu, Tania Vlad, Roxana Popescu, Cristian Vlad

**Affiliations:** 1Doctoral School “Engineering of Vegetable and Animal Resources”, University of Life Sciences “King Mihai I” from Timişoara, Calea Aradului 119, 300645 Timisoara, Romania; alexandru.nan@usvt.ro; 2Department of Biochemistry and Pharmacology, Faculty of Medicine, “Victor Babeș” University of Medicine and Pharmacy, 2nd Eftimie Murgu Square, 300041 Timisoara, Romania; dumitrascu.victor@umft.ro (V.D.); vlad.cristian@umft.ro (C.V.); 3Faculty of Bioengineering of Animal Resources, University of Life Sciences “King Mihai I” from Timisoara, Calea Aradului 119, 300645 Timisoara, Romania; 4ANAPATMOL Research Center, “Victor Babes” University of Medicine and Pharmacy, E. Murgu 2, 300041 Timisoara, Romania; puscasiu.daniela@umft.ro (D.P.); tania.vlad@umft.ro (T.V.); popescu.roxana@umft.ro (R.P.); 5Department of Cell and Molecular Biology, Faculty of Medicine, “Victor Babeș” University of Medicine and Pharmacy, 2nd Eftimie Murgu Square, 300041 Timisoara, Romania

**Keywords:** honey, polyphenols, cancer cell lines, antiproliferative effect, cell cycle arrest, apoptosis activation

## Abstract

Honey is a natural product which has been used throughout time as a food, spice, and medicine. Its therapeutic use has its origins in direct empirical observations of various beneficial actions in terms of its anti-infectious, anti-inflammatory, and wound-healing effects, to which an antiproliferative effect is added. In the context of malignant transformation, reductions in chronic inflammation, antioxidant action, cell cycle arrest, and apoptosis activation contribute to this antiproliferative effect, achievements attributed mainly to the polyphenols in its composition. A multitude of in vitro studies performed on malignant cell cultures try to elucidate the real mechanism(s) that can scientifically explain this action. In addition, its use as an adjuvant in association with cytostatic therapy demonstrates a promising effect in enhancing its cytotoxic effect, but also in reducing some adverse effects. Highlighting these actions allows for further perspectives to be opened regarding the use of honey for therapeutic and also prophylactic purposes, as a food supplement. Future studies will support the identification of real antiproliferative effects in patients with malignant tumors in terms of actions on the human body as a whole, moving from cell cultures to complex implications.

## 1. Introduction

Cancer is the cause of early death for 3 out of 10 people between the ages of 30 and 69 worldwide. A report by GLOBOCAN estimated that, by 2022, there will have been 20 million new cases of cancer and 10 million deaths worldwide [1]. Dietary changes and the consumption of raw plant foods with antiproliferative properties would reduce the risk of cancer. Processed and animal foods are on the list of pro-carcinogenic factors [2]. But, among these categories, there is a raw product somewhere in the middle, with antiproliferative properties: honey.

Since ancient times, honey has been used both as a sweetener and as a medicine for its various therapeutic properties. References to apitherapy have existed in Chinese, Greek, and Egyptian medicine, with references even in the Bible and the Quran [3]. The study of the tombs of Egyptian pharaohs has revealed that they used honey as medicine, food, and in various religious rituals [4]. In our times, honey is becoming an increasingly preferred product for its therapeutic properties.

Honey may differ depending on the geographical region it comes from and the type of flowers used in its production and collection. The differences in its properties may be due to different chemical compositions; these nuances are reflected in its properties, such as its antiproliferative and antitumor effects, and its healing of burns and other wounds [5].

Honey has been an important source of nutrients for both humans and bees. Collected nectar is transformed into honey through the bees’ glandular secretions, with bees being very selective when it comes to the type of plant they prefer in a given region, mainly focused on their nutritional needs [6]. Honey production can also be influenced by the stage of larvae, eggs, and the direct addition of pollen to honeycombs by beekeepers [7].

Many studies have demonstrated the antiproliferative, anti-infectious, anti-inflammatory, and healing-promoting properties of honey [8,9,10,11,12]. The antioxidant and healing-accelerating characteristics contribute to reducing the transformation of benign lesions into those with the potential for malignancy. These outcomes are attributed to the modulation of cytokine synthesis, and decrease oxidative stress, cytotoxic effects on cancer cells, and the acceleration of re-epithelialization, while preventing the multiplication of microorganisms. Many substances contained in honey are involved in this process, including flavonoids, coumarins, salicylates, polyphenols, minerals, and vitamins [13,14].

In this review, we bring to light the antiproliferative properties of honey in relation to its chemical composition and the abundance of certain constituents capable of modulating certain cellular functions. The intimate mechanisms of these effects have still not been fully elucidated; thus, the role of this review is to bring to the forefront what is known in the field, with further research coming in the future to complete and answer certain questions which are still unknown at present.

## 2. Methods

For this review, the selection of articles from the literature was made using the PubMed, Google Scholar, ScienceDirect, and Web of Science databases, with priority given to the most recent studies in the field to which contained certain information. Various search keywords were used, such as the following: “honey”, “antiproliferative effect”, “chronic inflammation”, “apoptosis”, “reactive oxygen species”, “antioxidant effect”, “Doxorubicin”, “5-Fluorouracil”, “Cyclophosphamide”, and “Tamoxifen”, as well as various combinations between them. Where additional related explanations were needed, the articles referring to that information were appropriately cited to offer high quality and easy understanding of the data provided.

## 3. The Abundance of Nutrients and the Chemical Composition of Honey

Chemical composition has a major influence on the clinical applications as well as the organoleptic properties of honey. Some of the most important compounds in the composition of honey are briefly discussed here.

### 3.1. Carbohydrates

Carbohydrates are the major component of honey, responsible for its specific, intensely sweet taste. Among them, we list fructose, the main component (41%), glucose (34%), and sucrose (1–2%) [15]. Here, the ratio between fructose and glucose is also important, which should be in the range of 0.9–1.35. A ratio of <1 indicates that the honey will undergo rapid crystallization, while a ratio > 1 characterizes a honey with slow crystallization [16]. This high sugar content creates a hyperosmolar environment unsuitable for the development of pathogenic microorganisms [17].

### 3.2. Aminoacids

Among the amino acids, proline is found in the highest amount, representing 50% of total amino acids [18], followed by serine, beta-alanine, glutamic acid, histidine, and glycine. The highest concentration was identified in multiflora honey (4866 nmol/L), probably essentially contributing to collagen biosynthesis [19]. In other studies, proline concentration was constant, independent of the type of honey studied [20].

### 3.3. Vitamins

Among the water-soluble vitamins, attention was paid to vitamins C, B1, B2 and B6, while the most studied fat-soluble vitamins were vitamin E and K [18,21]. A study has highlighted the production of hydroxyvitamin D3-type compounds in honey under the action of UV radiation originating from 7-dehydrocholesterol [22].

### 3.4. Polyphenols

Among the substances with antiproliferative capacity are polyphenols, responsible for honey’s antitumor effect and the reduction of carcinogenesis. They can be found as phenolic acids or in the form of flavonoids. The most important phenolic acids are gallic acid, para-coumaric acid, caffeic acid, chlorogenic acid, vanillic acid, vanillin (vanillic acid phenolic aldehyde), and syringic acid [23], which are displayed in Figure 1.

The number and location of phenolic OH groups contribute to antioxidant and free radical scavenging action [24], and, as can be seen from Figure 1, the compound with the highest number of phenolic OH groups is gallic acid, which, therefore, has the most intense antioxidant activity. Flavonoids, such as pinocembrin, luteolin, quercetin, kaempferol, chrysin, apigenin, and galangin (Figure 2), reduce oxidative stress, a process through which they can contribute to accelerating tissue regeneration, obviously also depending on the bioavailability of each individual compound [25].

## 4. The Main Mechanisms by Which Honey Components Produce Antiproliferative and Antineoplastic Action

There are multitudes of processes that honey interferes in, resulting in its antineoplastic activity. The most important ones, which have captured the attention of many researchers, are described below.

### 4.1. Stimulation of the Apoptosis of Cells with Structural Alterations

Honey has the ability to promote apoptosis through both intrinsic and extrinsic pathways. The effect of honey on apoptosis was demonstrated by the initiation of the intrinsic mitochondrial pathway, an effect attributed in particular to polyphenols through their stimulatory action on caspases 3, 7 and 9 [26] as well as via inhibition of anti-apoptotic proteins such as Bcl-2 and Bcl-xL [27]. Among these polyphenols, chrysin has been studied, which activates caspases 3 and 9 in tumor cells via Bak and Bax [28].

The extrinsic pathway of apoptosis is also influenced by honey. Honey appears to be able to modulate the epithelial mesenchymal transition (EMT), a process by which malignant cells invade, migrate, and then metastasize [28,29]. In cancer cells, E-cadherin expression is downregulated and N-cadherin is upregulated [29], a situation that Sangju honey was able to reverse, demonstrating an ability to induce E-cadherin upregulation and N-cadherin downregulation in human oral cancer cells treated with this type of honey [26]. The EMT process inhibits cancer cell apoptosis initiated by the death receptors TRAIL-R1 and TRAIL-R2, (tumor necrosis factor-related, apoptosis-inducing ligand receptor 1 and 2) members of the TNF family [29]. It seems that honey can also intervene by activating this process through TRAIL using different death receptors [30], chrysin demonstrating this capacity [14]. For example, the triggering of apoptosis in a hepatocellular carcinoma Hep G2 cell line treated with chestnut honey was achieved through the death receptor DR5, but also through the activation of Bax and caspases 3, 8, 9 [31]. This activation, both on intrinsic and extrinsic pathways, is demonstrated in vitro, but it remains to be studied what actual optimal quantity/day would produce, in humans suffering from malignant diseases, these activations of apoptosis with effects quantified in palpable parameters such as reduction in tumor size, regression of metastases, and stopping of the tumor invasion process.

### 4.2. Arresting the Cell Cycle of Tumor Cells

Another mechanism by which honey demonstrates its antiproliferative properties is its influence on the cell cycle of tumor cells, playing a role in inhibiting their uncontrolled multiplication. Antiproliferative activity was demonstrated by treating malignant cells with Saudi Sidr honey. Thus, the G0/G1 progression of colorectal cancer cells (HCT-116 cell line), breast cancer cells (MCF-7 cell line) and lung cancer cells (A-549 cell line) was prevented, arresting the G1 phase, but also shortening the S, G2 and M phases [32]. Honey also demonstrated cell cycle arrest in the G0 and G2/M phase after pancreatic cancer cell lines MIA PaCa-2 and AsPC-1 were exposed to honey for 24 h [33]. In this process, an important role is played by the expression of p53 as a tumor suppressor protein [34]. As is known, honey produces upregulation of p53 [34], which has a role in initiating apoptosis, but suppresses mutant genes of this protein, which are frequently found in tumor cells [33,35]. But the antiproliferative effect of honey was also associated with changes in cell cycle regulatory genes such as p21, p27, cyclins D1 and E as well as cyclin dependent kinases 2 and 4 with arrest in S and G2/M phases [36]. In this way, honey highlights its contribution in the first step towards cellular multiplication, preventing it in the case of neoplastic cells and creating the first defense barrier that can stop the uncontrolled multiplication of malignant cells.

### 4.3. Anti-Inflammatory Activity

Chronic inflammation occurring in different pathological contexts but that also develops during the aging process has an important contribution to malignant transformation through long-term induced tissue changes including angiogenesis, immunosuppression and alteration of the extracellular environment [37]. It seems that the effects of polyphenols on reducing TNFα-induced lesions, as well as in suppressing IL-1, IL-6, IL-8 and NF-kB expression, contribute to this process [36,38]. It has also been shown that honey has a valuable effect in inhibiting pro-inflammatory markers belonging to the metalloproteinase (MMP) class, especially MMP-2 and MMP-9. These markers showed increased values both in chronic inflammation and in malignant tumors, highlighting a cumulative anti-inflammatory and antitumor effect. It seems that the phenols in honey are responsible for this action [39].

The anti-inflammatory activity begins in the mouth, with oral administration, from the digestive tract, acting on the conditions that initiate and maintain local inflammation [40]. It is well known that oral lesions accompanied by chronic inflammatory process create an environment that promotes malignant transformation, maintained by the presence of Porphyromonas gingivalis, which is associated with chronic periodontal lesions [41]. In a study using a mucoadhesive film with honey applied directly to oral ulcers in forty-five Sprague–Dawley rats, a shortening of the healing period of the lesion was observed [42]. But, sometimes, other beneficial effects can also occur. For example, at the gastric level, chestnut honey rich in kynurenic acid did not demonstrate anti-inflammatory activity but prevented or improved lesions produced by indomethacin [43]. Below, at the intestinal level, honey with anti-inflammatory capacity has the ability to promote the production of short-chain fatty acids in the colon and to reduce the number of gram-negative bacteria [44,45]. When a model using fecal microbiota from the elderly was used, an increase in lactobacilli and a reduction in the number of gram-negative Enterobacteriaceae was observed, an effect attributed to the increased amount of gallic acid present in honey. This finding was positively correlated with IL-10 expression [44].

In the skin, it was observed that keratinocytes subjected to UVB radiation and then treated with honey showed a reduction in COX2 and NF-kB expression [46]. While UVB radiation is associated with malignant transformation through the changes it produces, we can say that the anti-inflammatory action of honey may play an important role in slowing or even stopping the skin carcinogenesis process.

### 4.4. Antioxidant Action

Even though anti-inflammatory processes contribute to antiproliferative mechanisms, there is a close correlation between inflammation and the generation of reactive oxygen species (ROS) due to the pro-inflammatory mediators produced. Polyphenols, and especially flavonoids, are responsible for this effect, produced through the synergism of the following three mechanisms: the scavenging of free radicals (i), increasing of the activity of enzymes involved in the removal of ROS, such as superoxide dismutase (SOD), glutathione reductase (GR), glutathione peroxidase (GP), and catalase (CAT) (ii), and the regulation of the expression of some genes involved in the defense against oxidative stress (iii) [47,48,49].

A large number of recent studies demonstrate the potential for free radical scavenging by initially determining the total phenolic content using the Folin–Ciocalteu reagent [50] and then quantifying the antioxidant activity with the method using the stable radical diphenyl-picrylhydrazyl (DPPH) [47,51,52,53,54,55]. This is due to the phenolic OH group(s) that can donate hydrogen atoms or electrons to inactivate ROS, stabilizing them and preventing the damage they cause [56]. Phenolic OH groups located in the ortho position (gallic acid, caffeic acid, chlorogenic acid), have the ability to chelate metal ions [56,57,58]. Moreover, caffeic acid has demonstrated a protective capacity on cell membranes, by inhibiting lipid peroxidation [56,59], and on DNA as well [59,60]. In the case of flavonoids, in addition to the phenolic OH groups, an important role is played by the double bond conjugated with an oxo group, favoring the delocalization of electrons and increasing the efficiency of the ROS removal and inactivation action [56,61].

Modulation of enzymatic activity against ROS has been demonstrated for flavonoids. Flavonoids, such as pinocembrin, showed a rise in SOD activity. In Wistar rats, pinocembrin has been shown to increase SOD activity in an experimental study in which the animals were exposed to CCl4. Administration of pinocembrin prior to exposure limited liver damage and reduced malondialdehyde (MDA) activity, a known marker of oxidative damage [62]. In C57BL/6J mice, pinocembrin from honey produced the same increase in SOD activity and a decrease in MDA under conditions of exposure to oxidative stress, in assays performed in hippocampus; thus, pinocembrin is seen to also have demonstrated a protective effect against oxidative stress [63]. Koc F et al. [64] showed that in mice with sepsis, chrysin caused an increase in SOD, CAT and GP activity, accompanied by a reduction in MDA (statistically significant values: *p* < 0.05). He also observed that the effects of chrysin were not dose dependent.

Modulating the expression of some genes may further contribute to the overall antioxidant activity of honey. In an in vitro experiment using lipopolysaccharide-activated Raw 264.7 murine macrophages, honey caused an increase in the expression of antioxidant factor genes heme oxygenase-1 (HO-1), thioredoxin reductase (TXNRD), and 4-nitroquinoline-N-oxide-1 (NQO-1), thereby inhibiting the production of pro-inflammatory factors [65].

Last but not least, it seems that the route of administration of the honey is also important. Thus, the digestion process may play an important role in the systemic effects of honey. In one study [66], it was shown that the antioxidant effect is more intense in the case of digested honey than before digestion. The overall antioxidant activity of honey is huge, but in the case of in vivo applications, the human body’s exposure to various and multiple oxidizing agents, their concentrations, and endogenous production under conditions of chronic stress should also be taken into account for an efficient therapeutic outcome.

## 5. Adverse Effects of Honey

In general, all compounds of natural or industrial origin have adverse effects, with the dose in which they are used being what makes the difference between the absence or presence of certain changes. Although the side effects of honey are not a very well-studied area, some of them have nevertheless been described.

The category of people with the most remaining questions is represented by patients with type 2 diabetes mellitus (DM2), because some studies indicate honey as a potentially beneficial sweetener in these patients [67], but other authors are reserved [68] or find this controversial [69]. In a study in which honey was administered in a quantity of 50g/day, divided into three doses, to patients with DM2, an increase in LDL-cholesterol levels and a reduction in adiponectin levels, an adipokine with anti-inflammatory and antiatherogenic effects, was found. The effect was attributed to the high fructose content, with the honey being of natural origin and with the suspicion of adulteration removed by the authors [70]. Other studies have shown an increase in HbA1c values at the same honey consumption of 50 g/day [71,72], but this increase is not evident at a consumption lower than 5–25 g honey/day [72].

In view of these arguments, in our opinion, honey consumption should be individualized for each patient, especially when we have people, often elderly, with multiple pathologies, such as cancer and DM2. In such a situation, a middle option should be chosen to maximize the antiproliferative effects of honey, while minimizing alterations to the lipid and carbohydrate profile as much as possible. From this discussion, we have excluded situations in which an allergy to one of the plant components that may be present in honey may occur, as well as the harmful and toxic effects of honey adulteration, because these are situations that are treated as allergic pathology or are considered toxicological issues.

## 6. Studies of the Effects of Honey on Different Cell Lines

Most studies on the mechanisms of action of honey have been conducted in vitro on various cell lines, especially cancer cell lines. In order to understand the compounds in the composition of honey that could have a therapeutic effect, these malignant cell lines have been subjected to various experimental conditions.

### 6.1. Epithelial Cell Lines

Dermatology represents a challenge regarding the therapeutic utility of honey, especially due to the possibility of its local application, directly to a lesion. Applying honey directly to a wound represents a therapeutic option for many chronic wounds, both in terms of reduced risks, minor adverse effects, and the absence of bacterial contamination. However, this has not been sufficiently explored for clinical application [73]. Some in vitro studies on skin cell lines have attempted to investigate what happens at the cellular level [74,75,76,77]. In one study, the role of honey against UVB radiation was examined in a post-irradiation HaCa Thuman keratinocyte cell line with and without pretreatment with honey. The cells that received pretreatment showed moderate damage, such as intracellular edema and a few sunburned cells, while those without pretreatment suffered complete necrosis [78]. In another study, the addition of multiflora honey and chestnut honey to HaCaT cells showed a significant increase of over 50% in cell viability after the addition of methylprednisolone [19]. This demonstrates the undeniable influence of honey in increasing cell viability and protecting against the aggression of physical and chemical agents, an essential phenomenon in preventing the initiation of malignant transformation.

### 6.2. Colorectal Cancer Cell Lines

Colorectal cancer is a type of cancer whose development is directly conditioned by dietary intake, including the type, quality, quantity, and contamination of food. Oral administration of honey as a dietary supplement could have a direct action on the mucosa, exerting its effects without requiring systemic absorption of biologically active compounds. A study conducted on the HCT-116 cell line demonstrated the antitumor effect of Saudi Sidr honey in a cytotoxic assay [32]. This property seems to be attributed to the synergistic action of flavonoids and phenolic acids. As some authors have shown [66,79,80], the effect of honey may also be conditioned by the digestion process undergone in the digestive tract until the site of action. The use of Manuka honey digested in vitro and applied to the HCT-116 cell line showed a dose-dependent anti-proliferative effect, the effect of which was more intense for the digested product (51%) compared to an undigested one (39%) [79]. The antiproliferative effect was demonstrated on the HCT-116 cell line and on the metastatic LoVo cell line by strawberry tree honey [81]. Strawberry tree honey, specifically, contains arbutin, a polyphenol with anti-inflammatory, antimicrobial, and antioxidant properties, as well as a tyrosinase inhibitor [82,83], which is considered a marker of strawberry tree honey [84]. Through its phenolic components, this honey reduced colony formation in a dose-dependent manner. In a study involving the HCT-116 cell line and on the metastatic LoVo cell line, the effect was attributed to the arrest of the cell cycle in the S and G2/M phases, as well as to the initiation of apoptosis via both the intrinsic and extrinsic pathways [81]. Although there are a multitude of types of honey, their antiproliferative effects on tumor cell lines have been experimentally demonstrated, with polyphenolic compounds playing a decisive role.

### 6.3. Breast Cancer Cell Lines

In vitro studies have shown a direct link between chronic inflammation and malignant transformation. One mechanism studied was the reduction of plasma IL-6 concentration; the phenomenon of increased IL-6 in chronic inflammation was frequently observed [85]. It has been shown that, through this mechanism, Manuka honey prevents the binding of IL-6 to the α chain of the IL-6 receptor. The compounds involved in this phenomenon are flavonoids, especially galangin, quercetin, luteolin, and chrysin [86]. The phenomenon has been demonstrated in the triple negative breast cancer cell line MDA-MG-231 and the non-small-cell lung cancer cell line A549 [86,87]. Human ERα positive MCF-7 and triple negative ERα/PR/HER2 MDA-MB-231 breast cancer cell lines showed a dose-dependent reduction in proliferation upon treatment with Manuka honey. Among the mechanisms implicated were the activation of AMP kinase and the inhibition of the mTOR and STAT3 pathways [26]. Although there are numerous studies presented in this review on the effect of honey on cancer cells, the question arises whether honey obtained directly from local producers has a different effect than that available commercially in stores. Karbasi S et al. [88] conducted a comparative study, between commercially available honey and Ziziphus jujube honey from beekeepers, on the effect of the different types of honey on the MCF-7 breast cancer cell line. Ziziphus jujube honey showed significantly superior antiproliferative effects in terms of slowing cell proliferation, reducing breast tumor cell viability, initiating apoptosis and the polyphenol content [88].

### 6.4. Lung Cancer Cell Lines

The effect of honey was also studied on human non-small-cell lung cancer (NSCLC) cell lines, such as H23 and A549, where Tualang honey demonstrated antitumor action by modulating the expression of proteins (88 proteins upregulated and 103 proteins downregulated for the H23 cell line; 66 proteins upregulated and 61 proteins downregulated for the A549 cell line) involved in the processes of angiogenesis, cell proliferation and apoptosis [89]. Another study conducted on the same H23 and A549 cell lines using Tualang honey also confirmed the same results, demonstrating the inhibition of malignant cell proliferation by arresting the cell cycle in the G1 and G2/M phases. In addition, promotion of apoptosis was observed through the intrinsic and extrinsic pathways by stimulating the expression of pro-apoptotic proteins (Bid, Bax, Caspase 3 and 8 and surface death receptor) and inhibiting the expression of anti-apoptotic proteins (Bcl-2 and Bcl-w) [90]. Moreover, Trigona itama honey produced cell cycle arrest in the G2/M phase of A549 cells after 72 h of exposure [91].

### 6.5. Pancreatic Cancer Cell Lines

Antitumor activity was demonstrated in the pancreatic cancer cell lines MIA PaCa-2 and AsPC-1. Exposure to honey for 24 h and then 48 h resulted in a reduction in growth and invasion compared to untreated cells [34]. In the pancreatic ductal adenocarcinoma cell line PANC-1, antitumor activity was attributed to caffeic acid, a polyphenol present in honey [92].

### 6.6. Prostate Cancer Cell Lines

Investigations of the therapeutic potential of honey in prostate cancer have been carried out on PC3 and DU145 prostate cancer cell lines, highlighting cytotoxic effects [93] but also the initiation of apoptosis via the extrinsic pathway where the effect was noted for kynureic acid from chestnut honey [94] or via the intrinsic pathway, the effect being studied on chrysin [95]. In the same tumor cells, Abel D.A.S. et al. [96] observed that honey polyphenols are able to inhibit the invasion process more intensely than the cell migration process. This effect is attributed to the inhibitory effect they have on metalloproteinase enzymes (MMPs), proteases expressed in metastatic cells that favor the lysis of cell adhesion [97], where honey seems to reduce the activity of MMP-2 and MMP-9 [39].

### 6.7. Hepatic Cancer Cell Lines

Several types of honey (manuka, arjuna, guggul, jiaogulan, olive) have been tested for their effects on oxidative stress in fatty acid-treated HepG2 liver cancer cell lines. They showed only partial evidence of a protective effect against oxidative stress [98]. The addition of Strawberry honey to the HepG2 cells did not demonstrate significant cytotoxic activity, and no substantial control of ROS [99]. If propolis and royal jelly are added to honey, the antioxidant and pro-apoptotic effect on the HepG2 cells is amplified. This effect was studied by comparing it with what happens to the normal liver cell line WRL-68. It was observed that the cytotoxic effect of antioxidants in the mixture was much more intense on HepG2 cells than on WRL-68 cells. Regarding apoptosis-inducing effects on liver cancer cells, the most intense effect was observed with chestnut honey with the addition of 10% royal jelly and 10% propolis [31]. Given these experimental findings, the following two conclusions arise: (i) for tumor liver cells, honey does not have the obvious antitumor effect of killing cells, an effect evident in other cell types, except when supplemented with propolis or royal jelly; (ii) if honey, instead of killing HepG2 cells, causes a transformation in their behavior from an invasive one to a non-invasive one, close to that of healthy cells, then additional investigations and detailed research in this direction are necessary.

### 6.8. Oral Cancer Cell Lines

The effects of honey are also visible on oral cancer. Oral cancer cell lines Ca9-22 and YD-10B that were treated with Sangju honey showed a reduction in epithelial mesenchymal transition through upregulation of E-cadherin and downregulation of N-cadherin. Honey treatment of these cells also initiated apoptosis through upregulation of p53 [34]. Oral squamous cell carcinoma induced by NQO-1 in Sprague–Dawley rats was studied after application of Tualang honey. A chemoprotective effect was observed when reducing the expression of genes involved in cell growth and cell cycle promotion, such as the epidermal growth factor receptor (EGFR) gene and the cyclin D1 gene, respectively, but also the expression of pro-inflammatory genes, such as COX-2 [100]. In light of these findings, honey could become an indispensable complementary agent in the treatment of malignant diseases.

In this sense, it is possible that the overall effect is not attributed to a specific compound(s), but rather to the synergistic effects that they may exert in the natural product. This phenomenon is quite common in natural extracts, where the effect of one substance is potentiated by the presence of others, even if they are minor components. An example of this would be the “entourage effect” in cannabinoids [101,102,103].

## 7. Studies on the Interaction Between Honey and Antineoplastic Treatment

Conventional antitumor treatment includes surgical resection of the tumor, and radiotherapy and chemotherapy administered in standardized regimens and adapted to each case in terms of the organ of interest, histological type, and stage of the disease. All these strategies are accompanied by severe adverse effects and often partial or temporary response, with relapses being observed in almost all neoplastic patients. There are recent trends to investigate the capacity of honey as an adjuvant in neoplastic patients, used together with some cytostatics.

### 7.1. 5-Fluorouracil (5FU)

5-Fluorouracil (5FU) is an antimetabolite cytostatic that is included in most chemotherapy regimens for advanced colorectal cancer [104,105,106,107], to which colorectal cancer cells have developed resistance [107,108,109,110,111,112]. To overcome this issue, the effect of honey in improving the therapeutic efficacy of 5FU has been studied, and several studies and their results are described below.

Manuka honey sensitizes HCT-116 cancer cells to 5FU in the initiation of apoptosis, which is important considering that resistance to apoptosis is one of the mechanisms of resistance to chemotherapy. This occurs by downregulating the following inhibitors of apoptosis: inhibitors of apoptosis proteins (IAPs), insulin-like growth factors (IGFs), and heat shock proteins (HSPs) [113]. Also, an important contribution is the downregulation of the transporter ATP-binding cassette (ABC) sub-family G member (ABCG2) [114], a transporter highly expressed in patients with chemotherapy resistance [115], an effect attributed to polyphenols [104]. In addition, the downregulation of c-myc with telomere shortening of colorectal cancer cells has an anti-metastatic effect [116]. Strawberry honey demonstrates the same effect on HCT-116 cells, as well as on the metastatic colon cancer cell line LoVo. Here, the process by which honey increases sensitivity to 5FU is mixed; it stops the cell cycle by decreasing EGFR expression, activates apoptosis via the intrinsic and extrinsic pathways, induces oxidative stress in cancer cells, and reduces the activity of MMP-2 and MMP-9 [117].

### 7.2. Doxorubicin (DOX)

Doxorubicin (DOX) is a chemotherapeutic agent belonging to the anthracycline class used in therapeutic regimens for the treatment of various malignant diseases, such as breast cancer, lymphomas, ovarian cancer, leukemias, and liver cancer [118,119,120,121]. The most important adverse reactions are represented by cardiotoxicity, nephrotoxicity, hepatotoxicity, neurotoxicity, and myelosuppression [122,123,124,125,126], where oxidative stress and inflammation play an essential role [127,128,129,130].

Application of Manuka honey to DOX-treated hepatocellular carcinoma cell lines HepG2 and Hep3P has been seen to result in a synergistic effect in dose-dependent activation of caspase 3 (4.3-fold increase) and the induction of apoptosis [131]. On the other hand, honey appears to be able to attenuate the adverse effects of DOX when administered to BALB/C mice, by way of decreasing liver enzymes, urea, creatinine, creatine kinase (CK), and lactate dehydrogenase (LDH) [132]. Honey-processed licorice was able to reduce the cardiac toxicity of DOX by inhibiting ROS production, increasing SOD activity, decreasing MDA, and preventing mitochondrial damage [133]. The same effects, in terms of reducing oxidative stress, were also produced in zebrafish by attenuating local myocardial inflammation, increasing SOD activity, and reducing MDA [134]. Another study specifically focused on the attenuation of DOX-induced nephrotoxicity in the context of honey usage also demonstrated reduced ROS generation, and decreased MDA and creatinine levels. Also, an adjuvant effect of honey during DOX treatment was identified in terms of enhanced apoptosis with caspase 3 activation and reduced Bcl-2 [135].

### 7.3. Cyclophosphamide (CY)

Cyclophosphamide (CY) is an agent used in combination therapy in the treatment of cancers, including various types of non-Hodgkin lymphoma, myeloma, ovarian adenocarcinoma, lung cancer, and breast cancer [136,137,138,139]. Honey may also represent a beneficial factor in association with this.

Algabi honey is an adjuvant for reducing CY toxicity by reducing immune suppression. In one study, the role of increasing serum IL-2, IL-6, and TNF-α levels in mice treated with CY is attributed to honey polysaccharides [140]. Under these conditions, polysaccharides can also cause dendritic cell maturation as well as increased secretory IgA production in the intestine [141]. During the administration of CY and Adriamycin to breast cancer patients, thyme honey demonstrated a reduction in oral mucositis and a decrease in its symptoms if the honey was used prophylactically [142]. Also, the association of Manuka honey proved to have anti-inflammatory, antioxidant, and anti-fibrotic protective effects against the development of hemorrhagic cystitis induced in rabbits treated with CY [143]. Honey produced by Apis dorsata bees, administered together with CY on A549 lung cancer cells, produced an increased cytotoxic effect on tumor cells. Among the mechanisms, stimulation of apoptosis with increased expression of the caspase 8 gene in parallel with downregulation of the Bcl-2 gene are involved. Its effects seem to be due to sphingolipids, especially those with sphingamine as the sphingoid base [144].

### 7.4. Tamoxifen

Tamoxifen is an anti-estrogen that is included in the cytostatic treatment regimens of patients with ER-positive breast cancer [145,146,147] to which malignant cells develop resistance in approximately 40% of patients [148]. In this regard, a series of reviews have been devoted to understanding the mechanisms by which this resistance develops [149,150,151,152,153]. Among the attempts to remedy this, honey is also included. Manuka honey enhances the antitumor activity of Tamoxifen on the MCF-7 breast cancer cell line [26], an effect attributed to the fact that polyphenols can function as phytoestrogens [154,155], sensitizing the cancer cell response [26].

The association of honey with cytostatic treatments opens up interesting perspectives regarding the amelioration of severe adverse effects as well as the potentiation of the antitumor cytotoxic effect. A hypothesis worthy of consideration is the possibility of the in vivo transformation of intensely proliferating cells into less invasive cells. Thus, the real benefits of honey will have to be studied in the future on large groups of patients to evaluate the situation in humans, much more complex than cell cultures or laboratory animals.

## 8. Conclusions

The effects of honey on tumor cells occur through several mechanisms of action, wherein polyphenols are the biologically active compounds that determine the majority of effects. Among these effects on cancer cells, attention has been drawn to the stimulation and promotion of apoptosis both in the intrinsic and extrinsic pathways, with the arrest of the cell cycle being closely correlated with apoptosis, as well as other actions, such as anti-inflammatory activity. The anti-inflammatory effect is obvious, starting from the site of administration at different areas of the digestive tract and going up to systemic effects. The antioxidant effect must also be taken into account, which is exerted both by increasing the activity of enzymes involved in the removal of ROS—the direct effect on ROS—and also by the regulation of the expression of some genes with a role in the neutralization of ROS. All these effects have been demonstrated on cell cultures, with each study trying to elucidate the intimate mechanism by which honey exerts its action. If we look at the different types of honey in comparison, we cannot say that one type is superior to another, because the chemical composition of natural products is dependent primarily on the type of plant(s) from which it is produced, but not least on the type of soil, climate, and weather conditions. In this sense, the composition may differ in the same geographical area from one year to another depending on the precipitation in the region, the transport of other substances by the wind, and also on possible pollution in the area. But we can say with certainty that natural honey is superior to synthetic honey or mixtures between natural honey and different synthetic constituents. With this in mind, the evaluation of different types of natural honey mixtures, with the idea of obtaining an effective formula customized for each type of cancer cell, would be interesting.

In addition, honey can be a promising adjuvant in the case of patients undergoing cytostatic treatment, as this association sometimes acts synergistically, accentuating the cytotoxic effect, while it sometimes contributes by reducing adverse effects. Thus, honey has demonstrated, through in vitro studies on cancer cell lines, that the association with 5FU not only enhances the antitumor effect, but is also able to reduce chemoresistance in treatment. It is interesting to observe the synergistic effect that the combinations of different types of honey and cytostatic drugs such as Manuka honey and 5FU, DOX, and Tamoxifen have. Another hypothesis worth considering is the possibility of honey being used to convert rapidly proliferating cells into cells with normal proliferative behavior. This idea of ours starts from the observation of the antimetastatic effect that honey can induce by regulating some genes that, at some point, modify their expression and favor tumor invasion. All these actions are studied mainly in vitro; studies that include large groups of patients are almost absent. This beginning of an in vitro study opens up broad perspectives for research on patients where concrete observations can be made, moving from the microcosm of cell cultures to the macrocosm of the organism as a whole. It also opens up the possibility of using honey as a dietary supplement in the context of pre-existing diseases, as well as for prophylactic purposes. We believe that honey appears as a complex food that could represent a valuable dietary supplement used both as an adjuvant together with certain standardized chemotherapeutic regimens, and also for prophylactic purposes in people at risk of developing malignant tumors. Future studies will have to elucidate aspects related to the real beneficial effects on the body, the amount of honey necessary to produce these effects, the optimal route and rate of administration, as well as interactions with cytostatic drugs.

## Figures and Tables

**Figure 1 nutrients-17-01595-f001:**
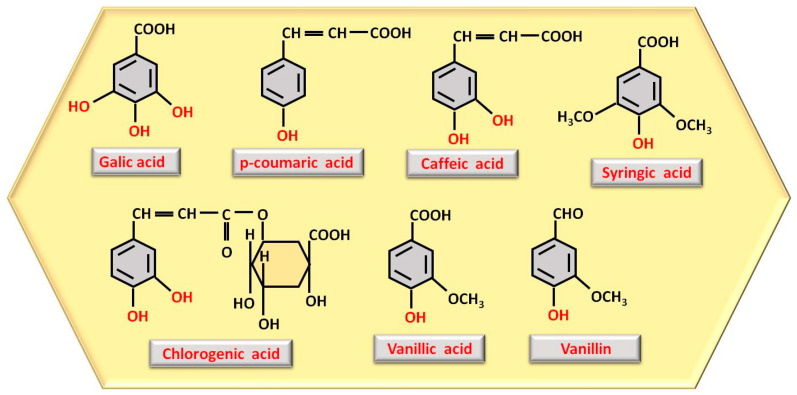
Structure representation of the most abundant polyphenols as phenolic acids observed in honey, with phenolic OH groups highlighted in red.

**Figure 2 nutrients-17-01595-f002:**
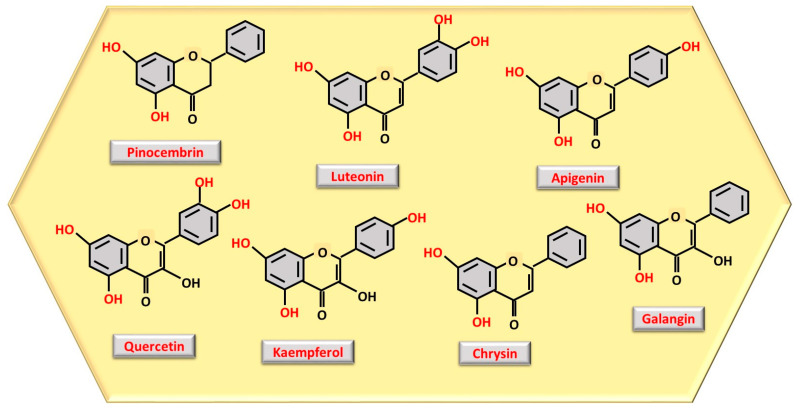
Structure representation of the most abundant polyphenols as flavonoids observed in honey, with phenolic OH groups highlighted in red.

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
