# Peer review of "From Chemical Composition to Antiproliferative Effects Through In Vitro Studies: Honey, an Ancient and Modern Hot Topic Remedy"

_nutrients, 2025, doi:10.3390/nu17091595_

Round 1

Reviewer 1 Report

Comments and Suggestions for Authors

This manuscript titled “ From chemical composition to antiproliferative effects through in vitro studies: honey, an ancient and a modern hot topic remedy ?”. The comments for this manuscript are as follows:

  1. It is well known that honey is a mixture. There are hundreds or even thousands of different kinds of honey in the world. I believe that most people have a positive view on the nutritional value of honey. But since it is a mixture, influenced by the geographical region and growing environment, its composition is believed to be diverse and different from year to year and from growth place. Just like red wine. Therefore, what the authors expressed in figure 2 is totally insufficient for scientific evidence. No mixtures can obtain a signal transduction result based on experiments. (That's meaningless) No body can answer who plays an important role in this mixture. Unless the authors perform a complete experiment with each pure compound, they are not qualified to show Figure 2. In addition, since honey is a mixture, the authors do not know whether other components may interact with each other, or even have side effects. Therefore, the reviewer strongly recommends that this kind of biased information be removed.
  2. In the text of manuscript, the authors only mentioned the advantages of honey and did not discuss the side effects. Such a review paper is not objective enough. Please strengthen and revise it. If honey is so good, then all departments in the hospital can be closed and everyone can just eat honey? Is it possible?
  3. There are many errors in the section of "Refererences". The writing of references should be consistent, and each word should not be capitalized. Please according to the "Instructions for Authors" to rewrite the references. For example: the format of Ref. 5, 8, 13, 16, 19, 20, 21… ect. Please revise it.

I decided it should be a minor revision.

Author Response

Comment 1: It is well known that honey is a mixture. There are hundreds or even thousands of different kinds of honey in the world. I believe that most people have a positive view on the nutritional value of honey. But since it is a mixture, influenced by the geographical region and growing environment, its composition is believed to be diverse and different from year to year and from growth place. Just like red wine. Therefore, what the authors expressed in figure 2 is totally insufficient for scientific evidence. No mixtures can obtain a signal transduction result based on experiments. (That's meaningless) No body can answer who plays an important role in this mixture. Unless the authors perform a complete experiment with each pure compound, they are not qualified to show Figure 2. In addition, since honey is a mixture, the authors do not know whether other components may interact with each other, or even have side effects. Therefore, the reviewer strongly recommends that this kind of biased information be removed.

Response 1: We removed figure 2 as requested by the referee. 

Comment 2: In the text of manuscript, the authors only mentioned the advantages of honey and did not discuss the side effects. Such a review paper is not objective enough. Please strengthen and revise it. If honey is so good, then all departments in the hospital can be closed and everyone can just eat honey? Is it possible?

Response 2: 

We introduced an extra-chapter dedicated to side effects of honey as suggested by the referee

5. Adverse effects of honey.

In general, all compounds of natural or industrial origin have adverse effects, the dose in which they are used being what makes the difference between the absence or presence of certain changes. Although the side effects of honey are not a very well-studied area, some of them have nevertheless been described.

The category of people with the most question marks is represented by patients with type 2 diabetes mellitus (DM2) because some studies indicate honey as a potentially beneficial sweetener in these patients [67] but other authors are reserved [68] or con-troversial [69]. In a study in which honey was administered in a quantity of 50g/day di-vided into three doses to patients with DM2, an increase in LDL-cholesterol levels and a reduction in adiponectin levels, an adipokine with anti-inflammatory and antiatherogenic effects, was found. The effect was attributed to the high fructose content, the honey being of natural origin with the suspicion of adulteration removed by the authors [70]. Other studies have shown an increase in HbA1c values at the same honey consumption of 50 g/day [71, 72] but this increase is not evident at consumption lower than 5-25 g honey/day [72].

In view of these arguments, in our opinion, honey consumption should be indivi-dualized for each patient, especially when we have people, often elderly, with multiple pathologies such as cancer and DM2. In such a situation, a middle option should be chosen to maximize the antiproliferative effects of honey but with minimizing alterations to the lipid and carbohydrate profile as possible. From this discussion, we have excluded situations in which an allergy may occur to one of the plant components that may be present in honey, as well as the harmful and toxic effects of honey adulteration, because these are situations that are treated as allergic pathology or belongs to toxicological issues.

Comment 3: There are many errors in the section of "Refererences". The writing of references should be consistent, and each word should not be capitalized. Please according to the "Instructions for Authors" to rewrite the references. For example: the format of Ref. 5, 8, 13, 16, 19, 20, 21… ect. Please revise it.

Response 3: We corrected the errors in the References chapter, removing the capitalization of each letter, as indicated by the referee.

Reviewer 2 Report

Comments and Suggestions for Authors

This review article is covering some very important aspects of chemical components of honey (polyphenols in flavonoids) and their antiproliferative effects capable of modulating certain cellular functions.   

The specific aims of this article are exclusively directed on multitude of in vitro studies of their various beneficial effects as anti-infectious, anti-inflammatory, reduction of chronic inflammation, antioxidant effect, cell cycle arrest and apoptosis activation. Effective highlighting of these beneficial effects will additionally deliver some further perspectives for therapeutic and prophylactic use of honey as a food supplement. All the presented data constituted the important goals and novelty of this paper.

The article is concluded with a collection of 145 mostly recent references. Additionally, 3 important figures are clear justification for the compilation of such review in specific chronological order.

The following suggested changes and recommendations should be introduced before the publication of the manuscript.

  1. Page 3. Line 112. Figure 1a should be renamed to figure 1 and figure 1b on page 4 to figure 2 and figure 2 on page 5 to figure 3. In that particular order the sequential chemical chronology will be preserved together with the mechanism activation/inhibition of apoptosis on figure 3. The name of P-coumaric acid on figure 1a should be corrected to p-coumaric acid.
  2. Page 4. Line 124. Change “compounds” with “components “.
  3. Page 4. Line 136 and 149. Change Figure 2 to figure 3 after correction of numbering of all figures.
  4. Page 6. Line 217. Insert the literature reference related to the Folin-Ciocalteu reagent.
  5. Page 7. Line 287. Insert the literature reference related to the strawberry tree honey effect.
  6. Page 11. Conclusion. This paragraph must be edited and expanded. In the present format, the authors do not fully describe the desired/anticipated effect and the final outcome of the paper. Authors should include comparative data of all selected types of honey, especially in their beneficial synergistic actions  For example Manuka honey enhances the antitumor activity of Tamoxifen on the MCF 7 breast cancer cell line and improving the therapeutic efficacy of 5FU. The additional information should be additionally documented with comparative literature references.

The manuscript is of good quality and importance, and is adequately written and edited in order to meet the standard for the articles published in Nutrients. Thus, I recommend it for publication after the correction of all these suggested minor changes and recommendations.

Author Response

Comment 1: Page 3. Line 112. Figure 1a should be renamed to figure 1 and figure 1b on page 4 to figure 2 and figure 2 on page 5 to figure 3. In that particular order the sequential chemical chronology will be preserved together with the mechanism activation/inhibition of apoptosis on figure 3. The name of P-coumaric acid on figure 1a should be corrected to p-coumaric acid.

Response 1: We changed the numbering of the figures, figure 1a became figure 1 and figure 1b became figure 2. We also made all the corresponding changes in the text as recommended by the referee 2. Also P-coumaric acid on figure 1 was changed to p-coumaric acid.

Comment 2: Page 4. Line 124. Change “compounds” with “components “.

Response 2: Page 4: we changed compounds to components

Comment 3: Page 4. Line 136 and 149. Change Figure 2 to figure 3 after correction of numbering of all figures.

Response 3: Page 4: we removed figure 2, which is a schematic representation of apoptosis and the places where honey could act, because reviewer 1 recommended its removal, so there's no need to change the numbering.

Comment 4: Page 6. Line 217. Insert the literature reference related to the Folin-Ciocalteu reagent.

Response 4: Page 6: we have inserted a citation about the Folin-Ciocalteu reagent (position 50 from references).

Comment 5: Page 7. Line 287. Insert the literature reference related to the strawberry tree honey effect.

Response 5: Page 7: a literature reference related to the strawberry tree honey effect was introduced together with the following comments: “ Strawberry tree honey particularly contains arbutin, a polyphenol with anti-inflammatory, antimicrobial and antioxidant properties as well as a tyrosinase inhibitor [82, 83], being considered a marker of strawberry tree honey [84].”

Comment 6: Page 11. Conclusion. This paragraph must be edited and expanded. In the present format, the authors do not fully describe the desired/anticipated effect and the final outcome of the paper. Authors should include comparative data of all selected types of honey, especially in their beneficial synergistic actions  For example Manuka honey enhances the antitumor activity of Tamoxifen on the MCF 7 breast cancer cell line and improving the therapeutic efficacy of 5FU. The additional information should be additionally documented with comparative literature references.

Response 6: The conclusion paragraph was expanded with some comments from authors’ perspective. In this regard, the following paragraphs have been introduced:

“If we look at the different types of honey in comparison, we cannot say that one type is superior to another because the chemical composition of natural products is dependent primarily on the type of plant(s) from which it is produced, but not least on the type of soil, climate and weather conditions. In this sense, the composition may differ in the same geographical area from one year to another depending on the precipitation in the region, the transport of other substances by the wind but also on possible pollution in the area. But we can say with certainty that natural honey is superior to synthetic honey or mixtures between natural honey and different synthetic constituents. Interesting from this point of view would be the evaluation of different types of natural honey mixtures until obtaining an effective formula customized for each type of cancer cell.

Thus, honey has demonstrated through in vitro studies on cancer cell lines that the association with 5FU not only enhances the antitumor effect but also is able to reduce chemoresistance to treatment. It is interesting to observe the synergistic effect that the combinations of different types of honey and cytostatic drugs such as Manuka honey and 5FU, DOX and Tamoxifen have. Another hypothesis worth considering is the possibility of honey to convert rapidly proliferating cells into cells with normal proliferative behavior. This idea of ours starts from the observation of the antimetastatic effect that honey can induce by regulating some genes that at some point modify their expression and favor tumor invasion.”

We believe that honey appears a complex food that could represent a valuable dietary supplement used both as an adjuvant together with certain standardized chemotherapeutic regimens but also for prophylactic purposes in people at risk of developing malignant tumors.”